# Investigation of the Electrical Coupling Effect for Monolithic 3-Dimensional Nonvolatile Memory Consisting of a Feedback Field-Effect Transistor Using TCAD

**DOI:** 10.3390/mi14101822

**Published:** 2023-09-23

**Authors:** Jong Hyeok Oh, Yun Seop Yu

**Affiliations:** ICT & Robotics Engineering, Semiconductor Convergence Engineering, AISPC Laboratory and IITC, Hankyong National University, 327 Jungang-ro, Anseong-si 17579, Gyenggi-do, Republic of Korea; rnjsdlr7@hknu.ac.kr

**Keywords:** monolithic 3-dimensional integration, nonvolatile memory, feedback field-effect transistor, electrical coupling

## Abstract

In this study, the electrical characteristics and electrical coupling effect for monolithic 3-dimensional nonvolatile memory consisting of a feedback field-effect transistor (M3D-NVM-FBFET) were investigated using technology computer-aided design. The M3D-NVM-FBFET consists of an N-type FBFET with an oxide–nitride–oxide layer and a metal–oxide–semiconductor FET (MOSFET) in the top and bottom tiers, respectively. For the memory simulation, the programming and erasing voltages were applied at 18 and −18 V for 1 μs, respectively. The memory window of the M3D-NVM-FBFET was 1.98 V. As the retention simulation was conducted for 10 years, the memory window decreased from 1.98 to 0.83 V. For the M3D-NVM-FBFET, the electrical coupling that occurs through an electrical signal in the bottom-tier transistor was investigated. As the thickness of the interlayer dielectric (*T_ILD_*) decreases from 100 to 10 nm, the change in the *V_TH_* increases from 0.16 to 0.87 V and from 0.15 to 0.84 V after the programming and erasing operations, respectively. M3D-NVM-FBFET circuits with a thin *T_ILD_* of 50 nm or less need to be designed considering electrical coupling.

## 1. Introduction

In recent decades, the scaling of transistors has continuously been improved, and the fabrication technology node of a metal–oxide–semiconductor field-effect transistor (MOSFET) has reached the nanoscale. However, the technical challenges of nanoscale transistors make it difficult to increase transistor integration [1]. Moreover, the 2-dimensional planar structure of the MOSFET has secondary effects such as a short-channel effect [2]. To overcome these limitations, various solutions have been suggested in terms of devices and circuit design [3,4,5,6,7,8,9,10]. Among them, monolithic 3-dimensional integration (M3DI), which allows the vertical stacking of transistors, logic gates, and memory devices, is a promising technology for future chip design [11,12,13,14,15]. Consequently, M3D structures have been researched in terms of transistors, logic gates, and system-level applications. For M3D structures, the transistor density within the same area is higher than that of conventional 2D structures. Moreover, the critical delay of M3DI is low due to vertical interconnection, which is shorter than horizontal interconnection [16]. In most of the configuration of the M3D structure, the logic gates are located in the lowest layer and the memory circuits and systems are designed in the upper layer [17,18,19]. To facilitate the stacking of each system, an investigation of the memory device, circuit, and system with a stackable structure is required.

One of the candidates for a next-generation memory device is a feedback field-effect transistor (FBFET). The FBFET exhibits an approximately zero slope and hysteresis characteristics [20,21,22]. Various circuits consisting of FBFETs operating as logic [23,24,25], memory circuits [26,27,28], and neuromorphic circuits [29,30] have been presented. In particular, the basic fabrication of the FBFET is based on the complementary metal–oxide–semiconductor (CMOS) technology; therefore, memory circuits configured with the FBFET have received attention as a next-generation memory system considering fabrication costs. Recently, a nonvolatile memory FBFET (NVM-FBFET) with a nanowire structure was suggested [31]. This NVM-FBFET demonstrates fast erasing and programming times (~1 μs) compared with conventional flash memory. Moreover, the suggested NVM-FBFET can operate as volatile memory (VM) using hysteresis characteristics. When using NVM-FBFET as volatile memory, the hysteresis characteristics themselves do not require a capacitor. This characteristic offers advantages for increasing the memory capacity through transistor scaling, whereas the conventional dynamic random access memory is challenged to increase density due to capacitor scaling [32]. When the VM-FBFET is designed with a M3D structure, data transmission is faster due to the reduced physical distance between NVM and VM. Additionally, the data bus width can be increased by stacking the memory circuits, as has already been realized by high-bandwidth memory [33]. However, the structure of the suggested NVM-FBFET makes it difficult to stack vertically. The nanowire structure can be designed as a 3D structure at the transistor level, but designing a 3D structure with circuits or at the system level is challenging. In order to stack the NVM-FBFET vertically for M3D design, an investigation of the stackable structure of the NVM-FBFET is required.

When designing the M3D structure, the electrical coupling between the top and bottom transistors or the monolithic interlayer via (MIV) occurs [34]. The electrical characteristics of the top-layer transistor were changed due to electrical coupling by the bottom-layer transistors or MIVs. The electrical coupling effects of the various circuits configured with MOSFETs [35], junction-less FETs [36,37], and FBFETs [38,39] have already been investigated. In the case of the NVM-FBFET, the investigation of the electrical coupling has not been conducted yet. For the M3D-NVM-FBFET, the current level considering the electrical coupling is an important factor that decides the ‘0’ and ‘1’ of a bit. Hence, the electrical coupling must be investigated before designing the M3D-NVM-FBFET.

In this paper, the electrical characteristics and the electrical coupling of the M3D-NVM-FBFET are investigated using technology computer-aided design (TCAD). First, the simulation structure of the M3D-NVM-FBFET will be explained in Section 2. Then, the electrical characteristics and the electrical coupling of the M3D-NVM-FBFET will be discussed in Section 3. Finally, the conclusions of this study will be described in Section 4.

## 2. Simulation Structure and Parameters

Figure 1a,b show the 3D schematic and its cross-section of the A–A’ of the M3D-NVM-FBFET, respectively. The NVM-FBFET located in the top tier in the M3D-NVM-FBFET was benchmarked from the published paper [31]. The bottom tier of the M3D-NVM-FBFET is one fully depleted silicon-on-insulator (FD-SOI) metal–oxide–semiconductor field-effect transistor (MOSFET). The M3D-NVM-FBFET consists of the N-type FBFET (NFBFET) including the tunneling oxide–nitride–blocking oxide (ONO) stack. The NFBFET structure is basically configured with p-n-p-n structure. For the M3D structure, an FD-SOI FET structure is used for the NFBFET. This optimal structure of the NFBFET has been researched already [38], and the structure was used for this M3D-NVM-FBFET. The M3D-NVM-FBFET can be fabricated based on the elaboration step of previous work [39], and deposition of the ONO layer must be added before deposition of the gate material in the fabrication flow [40]. The materials of the ONO stack are SiO_2_, Si_3_N_4_, and Al_2_O_3_, and the thickness of each ONO layer is 4, 5, and 6 nm, respectively. The total length of the channel region is 100 nm, and the length of each gated and ungated channel region is 50 nm. The doping concentration of each drain, source, and the ungated channel region is 1 × 10^20^ cm^−3^, and that of the gated channel region is 1 × 10^15^ cm^−3^. These structure parameters are described in Table 1. The simulation was conducted using commercial TCAD simulator Atlas [41]. The physical models including SRH, CVT, FERMI, BGN, CONMOB, FLDMOB, CONSRH, and AUGER for the FBFET simulation and PF.NITRIDE for the NVM simulation were used.

## 3. Simulation Results

In this section, the simulation results of the M3D-NVM-FBFET will be discussed. First, the mechanism of the M3D-NVM-FBFET will be explained in Section 3.1. Then, the memory characteristics of the M3D-NVM-FBFET for programming, erasing, and retention will be discussed in Section 3.2. Finally, the electrical coupling for the M3D-NVM-FBFET will be discussed in Section 3.3.

### 3.1. Mechanism of the M3D-NVM-FBFET

Figure 2a–c show the energy band of the off state, the forward sweep, and the on state of the M3D-NVM-FBFET, respectively. The black and red lines denote the valence band and the conduction band, respectively. Figure 3 shows the drain–source current (*I_DS_*)–gate–source voltage (*V_GS_*) characteristics of the M3D-NVM-FBFET with no charge in the nitride layer. For the off state of the M3D-NVM-FBFET, the drain–source voltage (*V_DS_*) and *V_GS_* are applied for 1 and −2 V, respectively. In this state, the carriers from the drain and source regions cannot inject into the channel region due to the potential barriers, as shown in Figure 2a. When the gate voltage starts the forward sweep at −2 V, the potential barrier in the gated channel region is lower, as shown in Figure 2b. Then, the electron from the source region can inject into the ungated channel region by thermionic emission. In the ungated channel region, the injected electrons accumulate and increase the carrier density. Subsequently, the potential barrier at the drain side is lower, so the hole from the drain region can diffuse into the ungated channel region. This positive feedback occurs through interaction between the electron and hole injection, as shown in Figure 2c. The positive feedback increases the current of the M3D-NVM-FBFET steeply, as shown in Figure 3. For the M3D-NVM-FBFET, the threshold voltage (*V_TH_*) is −0.43 V and the subthreshold swing is approximately zero.

### 3.2. Memory Characteristics of the M3D-NVM-FBFET

Figure 4a,b show the energy band of the ONO layer and the trapped electron concentration in the nitride layer for the programming and erasing operations, respectively. The black, red, and blue lines denote the valence band, the conduction band, and the trapped electron concentration, respectively. To simulate the programming operation, trap parameters used in the simulation are described in Table 2. Those parameters are basically based on fitted data on the fabricated NVM devices [42,43,44,45]. In order to avoid degradation of the memory performance such as the hot carrier effect [46,47], the very high voltage was applied to the gate electrode utilizing Fowler–Nordheim (FN) tunneling and the drain voltage for read operation is relatively low (~1 V). For the memory operation, the programming and erasing voltages are 18 and −18 V, respectively. The programming and erasing times are 1 μs. When the programming voltage is applied at the gate, the high electric field directs the potential of the tunneling oxide narrow in the field direction. Then, the electron from the silicon body can be transported into the nitride layer by FN tunneling and trapping in the nitride layer. Moreover, the trapped electron in tunneling oxide can be transported by Poole–Frenkel emission, as shown in Figure 4a. For the erasing operation, the negative voltage is applied at the gate, and then the direction of the electric field is reversed compared to the programming operation. The trapped electrons from the nitride layer are emitted and transported into the silicon channel region by FN tunneling, as shown in Figure 4b.

Figure 5a–c show the energy band of the off state, *I_DS_*–*V_GS_* characteristics of the M3D-NVM-FBFET after the programming and erasing operations, and *I_DS_*–*V_GS_* characteristics of the M3D-NVM-FBFET after five cycles, respectively. The red and black lines denote the results after the programming and erasing operations, respectively. For the programming operation, the trapped electrons accumulate holes in the gated channel region. Then, the fermi-energy level of the gated channel region is lower and the potential barrier is higher. Therefore, in order to turn to the on state, higher *V_GS_* is required. For the erasing operation, the trapped electrons are eliminated, and then the raised barrier is lower, as shown in Figure 5a. After the programming operation, the *V_TH_* of the M3D-NVM-FBFET shifts to 1.78 V. Then, after the erasing operation, the *V_TH_* shifts from 1.78 to −0.2 V. For the M3D-NVM-FBFET, the memory window is 1.98 V, as shown in Figure 5b. After the first cycle, the *V_TH_* after the programming operation shifts from 1.78 to 1.9 V, and stays almost the same after 2~5 cycles, but one was not changed after the erasing operation, as shown in Figure 5c.

Figure 6 shows the retention characteristics of the M3D-NVM-FBFET. The red and black lines denote the change in the VTH for the M3D-NVM-FBFET after the programming and erasing operations, respectively. The retention simulation was conducted for 10 years (3 × 10^8^ s). The memory window of the M3D-NVM-FBFET decreased from 1.98 to 0.83 V, as shown in Figure 6, so that it operates as an NVM. Finally, the comparison of the memory performance for various NVM-FBFETs is summarized in Table 3. The M3D-NVM-FBFET demonstrate the same performance for the programming and erasing operation, but improvements in read time and retention characteristics are needed when compared to other NVM-FBFETs. However, only the M3D-NVM-FBFET structure is designed considering M3DI.

### 3.3. The Electrical Coupling Effect for the M3D-NVM-FBFET

Figure 7 shows the *I_DS_*–*V_GS_* characteristics of the M3D-NVM-FBFET with no charge in the nitride layer at a thickness of the interlayer dielectric (*T_ILD_*) of 10 and 100 nm at a bottom-gate voltage (*V_BG_*) of 0 and 1 V. The black and red lines denote those of a *T_ILD_* of 10 and 100 nm, respectively, and the solid and dashed lines denote those of a *V_BG_* of 0 and 1 V, respectively. The electrical coupling for the M3D-NVM-FBFET is caused by the bottom-tier gate. When the *V_BG_* changes from 0 to 1 V, the *V_TH_* of the M3D-NVM-FBFET also changes, similar to the coupling effect of asymmetric double-gate FD-SOI [50] as follows:(1)$${\Delta V}_{TH}\approx \left(\frac{T_{ONO}}{T_{ILD}}\right)\left(\frac{\varepsilon_{ILD}}{\varepsilon_{ONO}}\right)\Delta V_{BG},$$
where Δ*V_TH_* represents the difference in the *V_TH_* when the *V_BG_* is changed by the Δ*V_BG_*, and *T_ONO_*, *ε_ILD_*, and *ε_ONO_* represent the thickness of the ONO stack and the permittivity of the ILD and the ONO stack, respectively. Here, 1/*ε_ONO_* = 1/*ε_blocking_* + 1/*ε_nitride_* + 1/*ε_tunneling_*, where *ε_blocking_*, *ε_nitride_*, and *ε_tunneling_* are the permittivity of blocking oxide, nitride, and tunneling oxide, respectively. When the *T_ILD_* decreases, the right-hand term of Equation (1) increases, then the Δ*V_TH_* increases, as shown in Figure 7. When the *T_ILD_* is 100 and 10 nm, the Δ*V_TH_* is 0.16 and 0.84 V, respectively.

Figure 8 shows the change in the *V_TH_* (Δ*V_TH_*) for the M3D-NVM-FBFET with various *T_ILD_* after programming and erasing operations, respectively. The solid and dashed lines denote the Δ*V_TH_* after the programming and erasing operations, respectively. The black, red, green, and blue lines denote the Δ*V_TH_* at *V_BG_* = 1, 0.8, 0.6, and 0.4 V, respectively. When the *V_BG_* is applied to the gate of the bottom transistor, the electric field is formed between the top and bottom transistors. Additionally, the magnitude of the field depends on the *T_ILD_* and the *V_BG_*. As the *T_ILD_* decreases and the *V_BG_* increases, the field is stronger, and the change in the *V_TH_* increases. When the *T_ILD_* decreases from 100 to 10 nm, the Δ*V_TH_* after the programming and erasing operations increases from 0.16 to 0.87 V and from 0.15 to 0.84 V, respectively. When the *V_BG_* increases at *T_ILD_* = 10 nm, the Δ*V_TH_* after the programming and erasing operations increases from 0.14 to 0.87 V and from 0.18 to 0.84 V. As the *T_ILD_* decreases, the Δ*V_TH_* increases abruptly at a *T_ILD_* of 50 nm. The electrical coupling must be considered below *T_ILD_* = 50 nm before designing the M3D-NVM-FBFET.

## 4. Conclusions

In this study, the electrical characteristics and electrical coupling effect of the M3D-NVM-FBFET were investigated using TCAD. The M3D-NVM-FBFET is configured with a NVM-FBFET with and a MOSFET in the top and bottom tiers, respectively. For the M3D structure, the FD-SOI structure was used for the NFBFET and the MOSFET. For the memory operation, the programming and erasing voltages were applied at 18 and −18 V, respectively, and the programming and erasing time was 1 μs. The memory window of the M3D-NVM-FBFET was 1.98 V. As the retention simulation was conducted over 10 years, the memory window decreased from 1.98 to 0.83 V, but it can be operated as an NVM. For the M3D-NVM-FBFET, the electrical coupling is caused by the electric field from the bottom-tier transistor, and the magnitude of the field depends on the *T_ILD_* and the *V_BG_*. When the *T_ILD_* decreases, the Δ*V_TH_* after the programming and erasing operations changes from 0.16 to 0.87 V and from 0.15 to 0.84 V, respectively. When the *V_BG_* increases at *T_ILD_* = 10 nm, the Δ*V_TH_* after the programming and erasing operations increases from 0.14 to 0.87 V and from 0.18 to 0.84 V. For the memory operation, the electrical coupling must be considered below *T_ILD_* = 50 nm before designing the M3D-NVM-FBFET.

## Figures and Tables

**Figure 1 micromachines-14-01822-f001:**
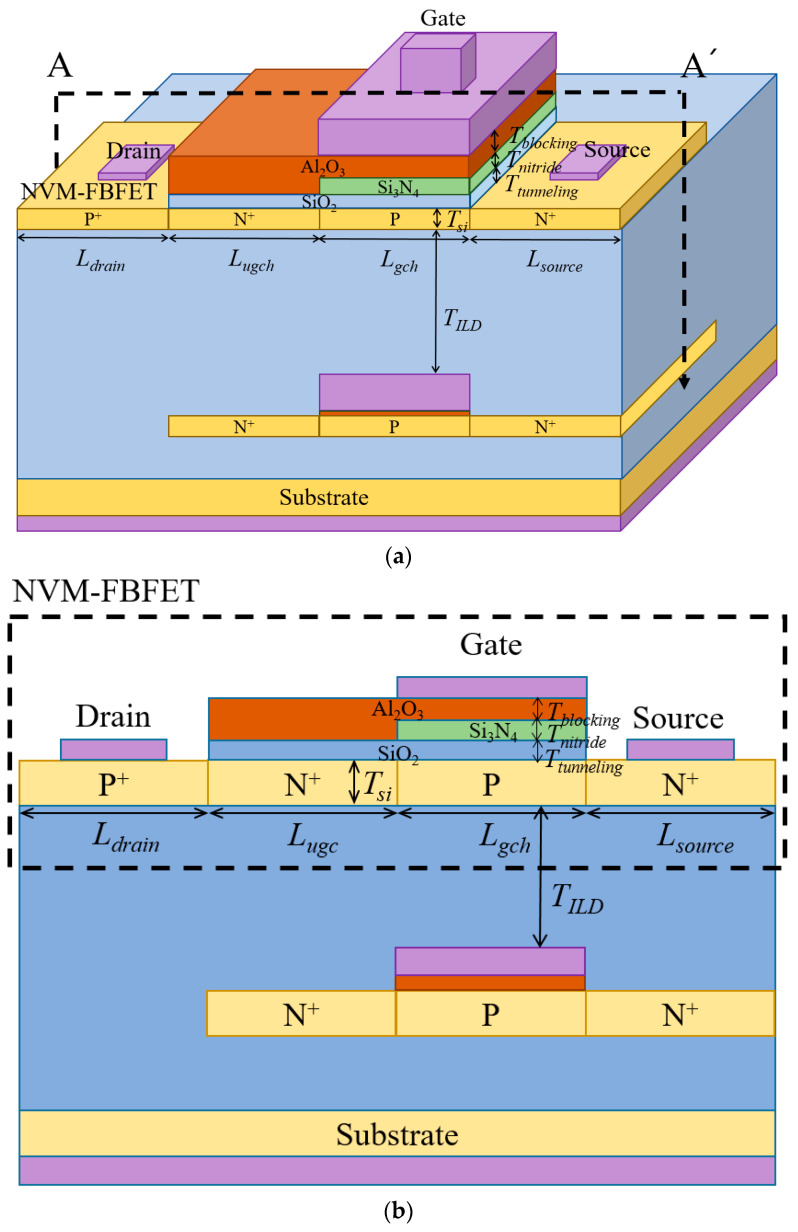
(**a**) A 3-dimensional schematic of the M3D-NVM-FBFET; (**b**) its cross-section of A–A’.

**Figure 2 micromachines-14-01822-f002:**
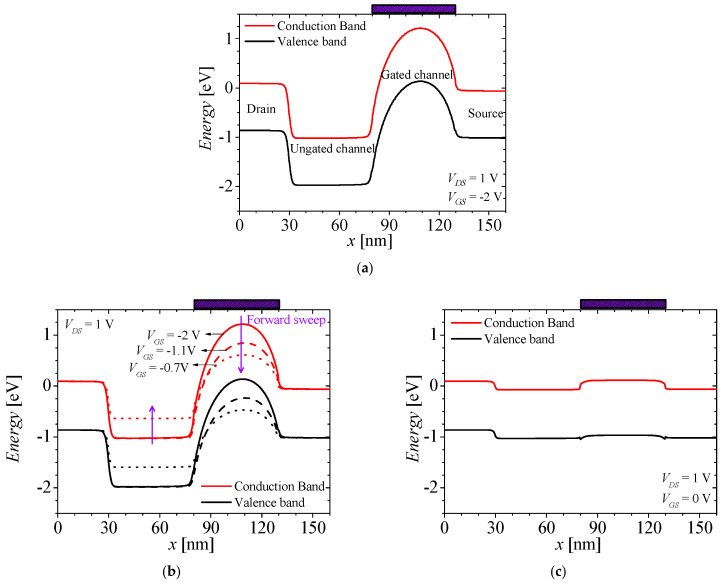
The energy band of the M3D-NVM-FBFET for (**a**) the off state (*V_DS_* = 1 V and *V_GS_* = −2 V), (**b**) the forward sweep at different *V_GS_* (*V_DS_* = 1 V), and (**c**) the on state (*V_DS_* = 1 V and *V_GS_* = 0 V).

**Figure 3 micromachines-14-01822-f003:**
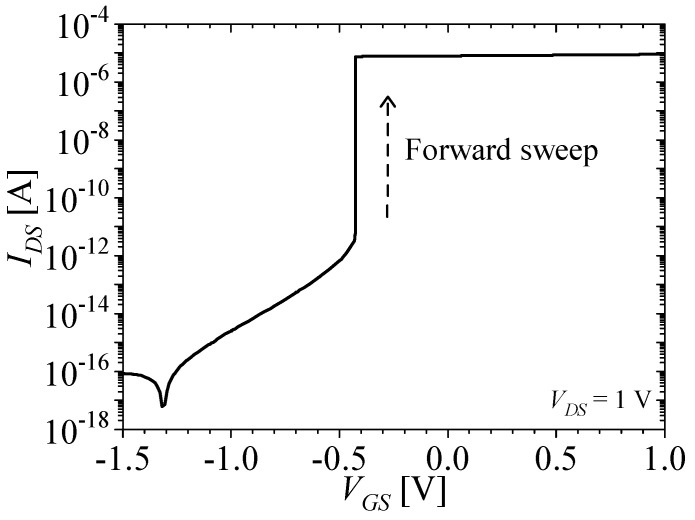
*I_DS_*–*V_GS_* characteristics of the M3D-NVM-FBFET with no charge in the nitride layer (*V_DS_* = 1 V).

**Figure 4 micromachines-14-01822-f004:**
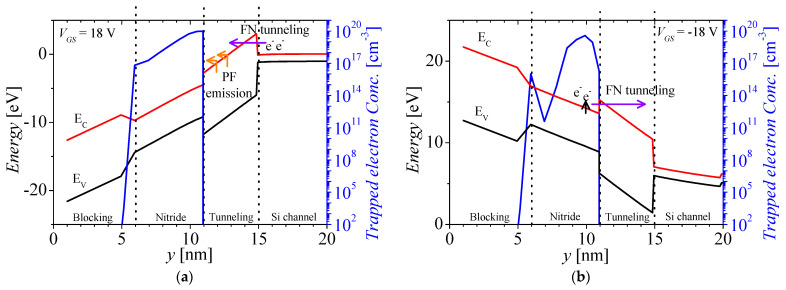
Energy band of the ONO layer and the trapped electron concentration in the nitride layer for (**a**) programming (*V_GS_* = 18 V) and (**b**) erasing operations (*V_GS_* = −18 V).

**Figure 5 micromachines-14-01822-f005:**
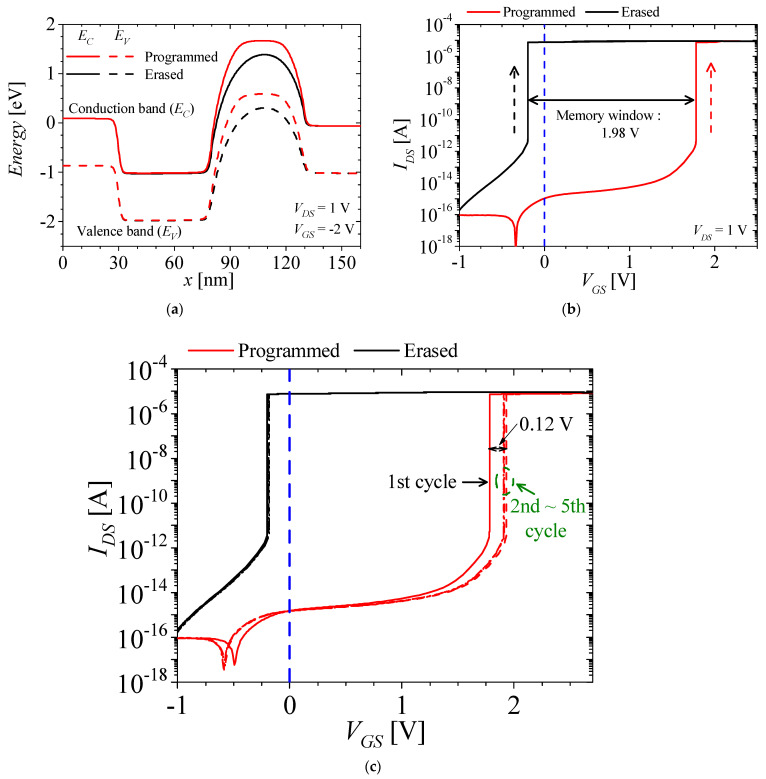
(**a**) Energy band in the off state (*V_DS_* = 1 V and *V_GS_* = −2 V), (**b**) *I_DS_*–*V_GS_* characteristics of the M3D-NVM-FBFET (*V_DS_* = 1 V) after the programming operation (red line) and the erasing operation (black line), and (**c**) *I_DS_*–*V_GS_* characteristics of the M3D-NVM-FBFET after five cycles.

**Figure 6 micromachines-14-01822-f006:**
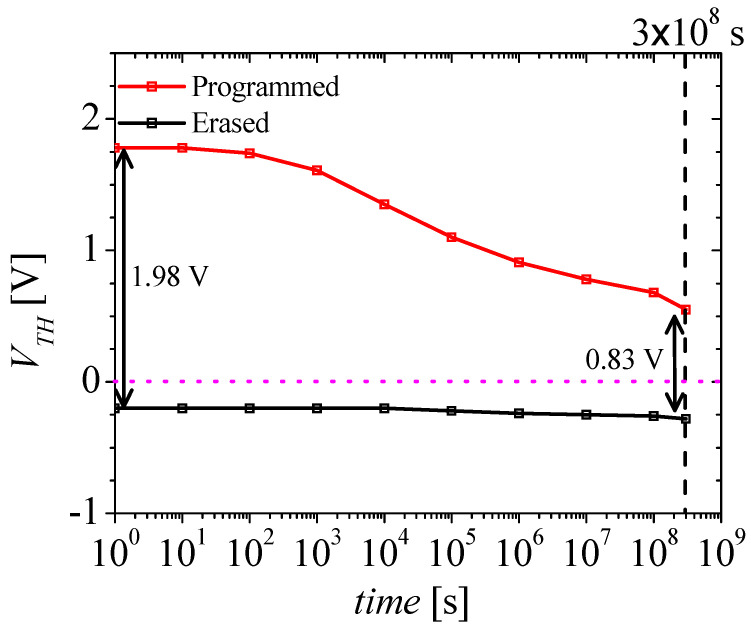
Retention characteristics of the M3D-NVM-FBFET for 10 years.

**Figure 7 micromachines-14-01822-f007:**
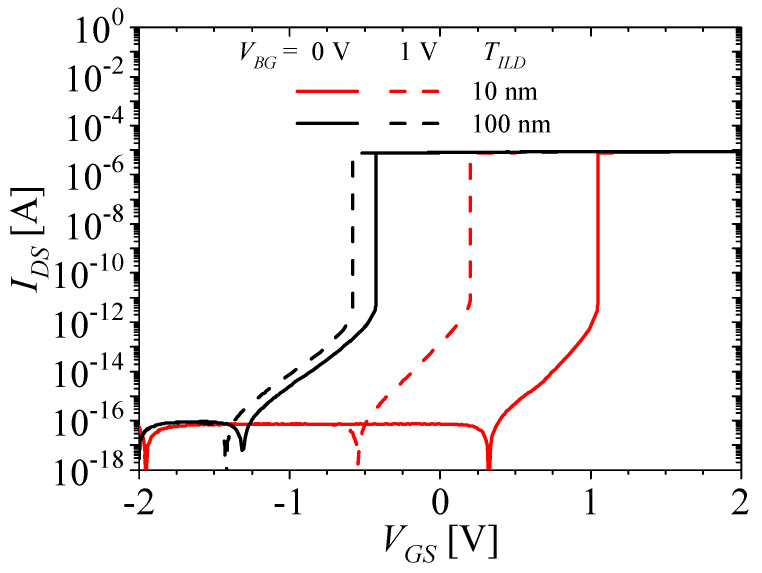
*I_DS_*–*V_GS_* characteristics of the M3D-NVM-FBFET with no charge in the nitride layer at *T_ILD_* = 10 and 100 nm at *V_BG_* = 0 and 1 V.

**Figure 8 micromachines-14-01822-f008:**
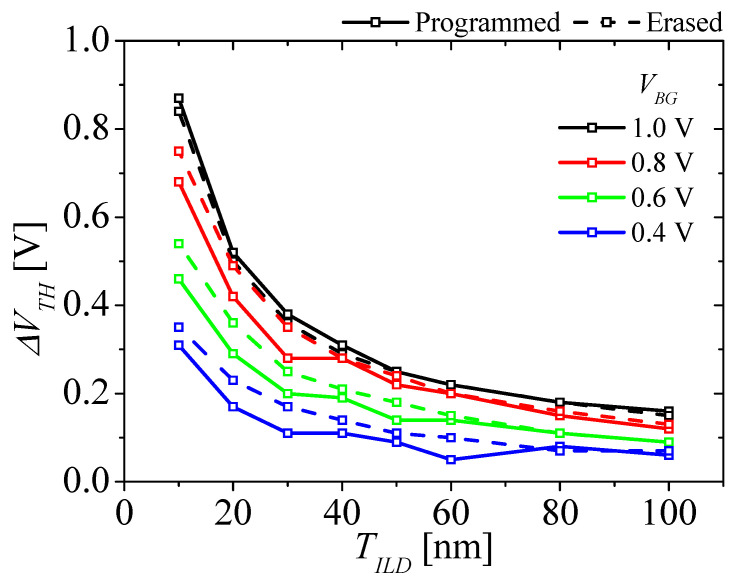
Change in the *V_TH_* for the M3D-NVM-FBFET with various *T_ILD_* after the programming (solid line) and erasing (dashed line) operations as a *V_BG_* of for 1.0 (black), 0.8 (red), 0.6 (green), and 0.4 V (blue) is applied.

**Table 1 micromachines-14-01822-t001:** Structure parameters of the M3D-NVM-FBFET.

Parameters	Description	Value/Unit
*L_drain_*, *L_source_*	Length of the drain and source regions	30 nm
*L_ugch_*	Length of the ungated channel region	50 nm
*L_gch_*	Length of the gated channel region	50 nm
*T_blocking_*	Thickness of the blocking oxide layer (Al_2_O_3_)	6 nm
*T_nitride_*	Thickness of the nitride layer (Si_3_N_4_)	5 nm
*T_tunneling_*	Thickness of the tunneling oxide layer (SiO_2_)	4 nm
*T_si_*	Thickness of the silicon body	6 nm
*T_ILD_*	Thickness of the interlayer dielectric (ILD)	Var.
*N_drain_*, *N_source_*	Doping concentration of the drain and source regions	1 × 10^20^ cm^−3^
*N_ugch_*	Doping concentration of the ungated channel region	1 × 10^20^ cm^−3^
*N_gch_*	Doping concentration of the gated channel region	1 × 10^15^ cm^−3^
*Φ_FBFET_*	Work function of the gate metal	5.3 eV

**Table 2 micromachines-14-01822-t002:** Trap parameters for the nitride layer for the M3D-NVM-FBFET.

Parameter	Value/Unit
Trap energy level	1.2 eV
Effective mass electron, *m_e_**	0.33 *m*_0_
Effective mass hole, *m_h_**	0.46 *m*_0_
Capture cross-section	1 × 10^−13^ cm^2^
Trap density	1 × 10^20^ cm^−3^

**Table 3 micromachines-14-01822-t003:** Comparison of the memory performance for various NVM-FBFETs.

NVM Devices	Program Time	Erase Time	Read Time	Monolithic 3D Integration
NOR Flash [48]	1 μs	1 ms	15 ns	X
NW NVM-FBFET [31]	1 μs	1 μs	200 ns	X
FinFET NVM-FBFET [49]	1 μs	1 μs	10 ns	X
This work	1 μs	1 μs	~100 ns	O

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
