# Peer review of "Investigation of the Electrical Coupling Effect for Monolithic 3-Dimensional Nonvolatile Memory Consisting of a Feedback Field-Effect Transistor Using TCAD"

_micromachines, 2023, doi:10.3390/mi14101822_

Round 1

Reviewer 1 Report

The comments and suggestions are given below the paper titled: Electrical Coupling Effect for Monolithic 3-Dimensional Non- 2 volatile Memory Consisting of Feedback Field Effect Transistor

In the present work, the electrical characteristics and electrical coupling  effect  for monolithic  dimensional nonvolatile memory was investigated using numerical simulations.  The paper is well written. Moreover, this work investigated an interesting topic regarding memory cells. However, some points should be highlighted by the authors as given below:  

• On first observation is that the paper is mainly proposed a design for memory applications, with a lack of comparison with published works regarding the development of memory cells. In this regard, I suggest introducing a table of comparison to show the advantages of the proposed design in comparison to that of the published works based on deferent structures and material strategies.

• The authors used a single gate FET for designing the proposed memory cell, I think this structure exhibits degraded performances caused by the short channel effects (SCEs). In this context, new emerged structures such Multi-gate FETs, TFET,.. were proposed for high performance ULSI memory cells, I suggest introducing some sentences, explanations and perspectives  regarding this point . For more details, please refer to:  https://doi.org/10.1016/j.optlastec.2017.06.002; DOI 10.1088/2053-1591/aac756; https://doi.org/10.1016/j.nima.2012.04.072.

 • The interfacial traps and hot-carriers effects (HCEs) can greatly affect the performance and degradation behavior of the proposed design, I suggest discussing this point. To do so, please refer to: DOI 10.1088/1674-4926/33/1/014001; https://doi.org/10.1016/j.microrel.2010.10.002; https://doi.org/10.1002/pssc.201200128

• The proposed structure is a complex design, I think  some sentences regarding its elaboration steps should be discussed. 

• Several language mistakes should be corrected. 

In summary, the achieved work showcases significant contributions and provides pertinent insights on the development of memory cells. Thus, I would recommend with Major revision this manuscript for the possible publication in the Journal.

Some language mistakes should be corrected. 

Author Response

Please find an answer to the Reviewers' comments and a revised manuscript.

sincerely,

Yun Seop Yu

Reviewer 2 Report

Jong Hyeok, et al., demonstrated the electrical coupling effect for monolithic 3D non-volatile memoru consisting of feedback FET. The topic of the research is at the interest of the readership of the journal. The manuscript can be accepted for publication after addressing minor querries below.

1. Since authors study is relying on simulation, they can choose some range for the materials paramenters in the device such as thickness, length of the channel, doping concetration, etc. This could be provide much wider idea for the readers intend to perform similar experimental studies.

2. What would be the best way ot the tuning parameter to acheive lower erasing voltages for the proposed device?

3. Authors observed that the programing and erasing operations increases from 0.15 to 0.84 range. Is it a continuous process. How many cycles have you observed and what would be the overall change?

No further language editing is required. 

Author Response

Please find an answer to the Reviewers' comments and a revised manuscript.

Sincerely,

Yun Seop Yu

Reviewer 3 Report

Yun Seop Yu and the coauthors presented a simulation study on the electrical coupling effect for monolithic 3-dimensional non-volatile memory consisting of the feedback field effect transistor. The working mechanism, memory characteristics, and the coupling effect were simulated. This work can be further improved and accepted with some revisions. The suggestions are listed as follows:

1.      The title should revised to make it clear to the reader that this is a work based on computational and simulation.

2.      On line 29, for “various solution”, more example should be given and compared.

3.      In the introduction part, related simulation works should be added besides the experimental ones.

4.      The words are unreadable in Fig 1.

5.      In Part 3, the basic mechanism, memory characteristics seem clear. The 3.3 Electrical coupling part is weak, which should be the key part, as indicated in the title. More related details, discussion and investigation should be provided. For example, a continuous change of Vbg should be added in Figure 7. How do non-volatile memory and feedback properties go with the Vbg?

The language is fluent, and could be further improved with grammar and spelling checks.

Author Response

(The authors gave the same response as above.)

Round 2

Reviewer 3 Report

The authors have addressed all my concerns, and it can be published in this form.